# Tooth Loss and Blood Pressure in Parkinson’s Disease Patients: An Exploratory Study on NHANES Data

**DOI:** 10.3390/ijerph18095032

**Published:** 2021-05-10

**Authors:** Patrícia Lyra, Vanessa Machado, Luís Proença, José João Mendes, João Botelho

**Affiliations:** 1Centro de Investigação Interdisciplinar Egas Moniz (CiiEM), Clinical Research Unit (CRU), Egas Moniz—Cooperativa de Ensino Superior, 2829-511 Caparica, Portugal; patricialyra10@gmail.com (P.L.); vmachado@egasmoniz.edu.pt (V.M.); jmendes@egasmoniz.edu.pt (J.J.M.); 2Evidence-Based Hub, CRU, CiiEM, Egas Moniz—Cooperativa de Ensino Superior, 2829-511 Caparica, Portugal; lproenca@egasmoniz.edu.pt; 3Quantitative Methods for Health Research (MQIS), CiiEM, Egas Moniz—Cooperativa de Ensino Superior, 2829-511 Caparica, Portugal

**Keywords:** Parkinson’s disease, oral health, tooth loss, edentulism, hypertension, diabetes mellitus

## Abstract

**Objectives:** To evaluate tooth loss severity in PD patients and the impact of missing teeth on blood pressure (BP) and glycated hemoglobin (Hba1c) levels. **Methods:** All adults reporting specific PD medication regimens with complete dental examinations were included from the NHANES 2001 to 2018 databases. Sociodemographic, systolic BP (SBP), diastolic BP (DBP) and Hba1c data were compared according to tooth loss severity, and linear regression analyses on the impact of tooth loss on SBP, DBP and Hba1c levels were conducted. **Results:** The 214 included participants presented 9.7 missing teeth, 23.8% severe tooth loss and 18.2% total edentulousness. Severe tooth loss cases were significantly older (*p* < 0.001), had higher smoking prevalence (*p* = 0.008), chronic medical conditions (*p* = 0.012) and higher Hba1c (*p* = 0.001), SBP (*p* = 0.015) and DBP (*p* < 0.001) levels. Crude and adjusted linear models revealed a relationship between SBP, DBP and missing teeth; however, age confounded these links (SBP: B = 0.10, SE = 0.16, *p* < 0.05; DBP: B = 0.16, SE = 0.10, *p* < 0.05). Tooth loss presented no significant relationship with Hba1c levels. **Conclusions:** Severe tooth loss is prevalent among PD patients. Blood pressure levels showed a positive linear relationship with the number of missing teeth, although age was a confounding factor. Furthermore, tooth loss and Hba1c levels revealed no significant linear relationship.

## 1. Introduction

Parkinson’s disease (PD) is a slowly progressive and over time disabling condition of the nervous system, affecting around 680,000 US citizens and less than 1% of North Americans over 45 years of age [1,2]. Although its etiology remains elusive to date, a complex interaction of environmental factors with genetic susceptibility is currently the most accepted definition [3]. Clinical PD is heterogeneous, with resting tremor, muscular rigidity and bradykinesia as the classical motor features [4,5]. However, complications might develop, such as gait abnormality, loss of balance, autonomic dysfunctions, speech impairments and cognitive decline, oftentimes leading to dementia [6]. Hence, PD patients’ overall quality of life can be heavily affected as everyday life tasks (i.e., personal hygiene routines) become an increasing challenge, especially as treatment difficulties arise in later disease stages and patient disability aggravates [7].

The oral health condition of PD patients tends to be deteriorated due to the motor and cognitive dysfunction this neurodegenerative condition presents [8,9]. As a result, individuals with PD have deficient oral hygiene which precipitates the development of periodontitis (a chronic inflammatory condition of the supporting structure of the teeth) [10] and carious lesions, and consequently retained roots [11] and tooth loss [12]. Furthermore, PD patients may have masticatory dysfunction and can develop dysphagia, chewing problems and levodopa-induced orofacial dyskinesia and dystonia of the tongue and masticatory muscles, alongside impaired manual dexterity [13,14]. Still, further research is needed on the true extent of the oral health repercussions of PD.

From a systemic point of view, these unbalanced oral hygiene habits, with a tendency to aggravate as PD progresses, may have negative consequences. On the one hand, our group has demonstrated that PD cases with periodontitis are associated with leukocytosis [15], although the link between other oral conditions with other systemic markers (i.e., blood pressure (BP) or glycated hemoglobin (Hba1c)) is still seldom. High BP, which burdens about 45% of the US adult population, is associated with cognitive decline and premature mortality rates [16,17,18], and not only has been linked to periodontitis [19,20] and tooth loss [21,22,23,24], but blood pressure fluctuations have also been associated with the autonomic dysfunctions in PD [25]. Additionally, diabetes mellitus, affecting approximately 13% of US adults [26], whose cardiovascular, renal, ocular and neuronal complications are well established in literature [27], also has a known bidirectional relationship with severe periodontitis [28] and is a possible risk factor for PD, although further research is needed to clarify the biological mechanisms of such an association [29]. However, the link between BP levels (or Hba1c) and tooth loss count in PD cases has never been studied, although a prospective positive link may arise according to several recent lines of evidence [18,20,22,30,31].

In this study, our primary aim was to assess tooth loss severity in individuals with PD. Secondly, we evaluated whether missing teeth is associated with BP and Hba1c levels.

## 2. Materials and Methods

### 2.1. Study Design and Participants

Datasets from 2001–2002 to 2017–2018 National Health and Nutrition Examination Survey (NHANES) were collected and then analyzed in this secondary study. The NHANES is a stratified multistage survey aiming to representatively assess the non-institutionalized U.S. population health status. All data were collected through interviews, physical exams and laboratory exams. Furthermore, all waves of NHANES were reviewed and approved by the Centers for Disease Control (CDC) and Prevention National Center for Health Statistics Research (NCHS) Research Ethics Review Board (protocols #98-12, #2005-06, #2011-17 and #2018-01, Atlanta, GA, USA), and all study participants provided written informed consent [32]. Further information on the sampling process, methodological design, gathering of medical records and dental data collections can be looked up at www.cdc.gov/nchs/nhanes.htm (accessed on 15 April 2021).

For the purpose of the present study, the inclusion criteria were as follows: participants who underwent dental examinations and who reported medication regimens indicative of PD. The exclusion criteria were defined as follows: unsecure PD medication as previously defined (Cabergoline, Orphenadrine and Pramipexole) [15]; participants younger than 18 years of age; and individuals with missing data on the study variables. This study followed the Strengthening the Reporting of Observational Studies in Epidemiology (STROBE) guideline [33] (Appendix A).

### 2.2. Variables and Data Measurement

#### 2.2.1. PD Definition

The identification of PD cases was conducted through self-reported PD medications in the NHANES database, according to a previous study methodology [15]. In this sense, PD cases were defined if patients reported the use of the following secure PD medications: Benztropine, Carbidopa, Levodopa, Ropinirole, Methyldopa, Entacapone and Amantadine [15].

#### 2.2.2. Dentition Clinical Examination

Upon collection of Oral Health Dentition Examination Data from the NHANES in the years 2001–2002, 2003–2004, 2009–2010, 2011–2012, 2013–2014, 2015–2016 and 2017–2018 and the Oral Health Examination Data from the NHANES in 2005–2006 and 2007–2008. Third molars were excluded from the analysis. Thus, the following dentition-related variables were computed: total number of teeth present, total number of missing teeth, number of implants, number of molars, premolars and canines. Tooth loss was categorized as “severe” if 10 or more teeth were lost, and “non-severe” if less than 10 teeth were lost [34].

#### 2.2.3. Health Characteristics

In order to assess the systemic health status of the study participants, the total number of chronic medical conditions was considered as a continuous variable for the purpose of data handling and was reached upon analysis of self-report of asthma, psoriasis, gout, congestive heart failure, coronary heart disease, angina, heart attack, stroke, emphysema, thyroid, bronchitis, liver disease and cancer. Diabetes mellitus (DM) and hypertension were separately appraised variables due to their known strong impact on oral health [35,36,37,38]. DM was defined through a previous report and confirmed with Hba1c values (>6.5%) [39]. Uncontrolled DM cases were defined through Hba1c >8% [40]. Hypertension was defined from a past history of informed hypertension (self-reported) and/or systolic BP (SBP) >140 mmHg or diastolic BP (DBP) >90 mmHg [41].

#### 2.2.4. Covariates

Self-reported information about sociodemographic and lifestyle parameters regarding age; sex; race/ethnicity (i.e., Mexican American, Non-Hispanic White, Non-Hispanic Black, other Hispanic and other race—including multi-racial); educational level; marital status (i.e., single, married/living with a partner, and divorced/separated/widowed [42]); family income-to-poverty ratio; and smoking status were collected. Educational level was categorized as “<high school” (including less than 9th grade and 9–11th grade, which includes 12th grade with no diploma), “high school” (including high school grad/GED or equivalent) and “>high school” (including some college or AA degree and college graduate or above) [43]. The reported family income-to-poverty ratio is ranked continuously from 0 to 5, with “0” corresponding to “no income” and “5.0” corresponding to an income 5 or more times above the federal poverty threshold (FPL) [44]. Smoking status was categorized based on a self-reported questionnaire where: active smokers reported a consumption of ≥100 cigarettes during their lifetime and were still currently smoking; former smokers reported smoking ≥100 cigarettes during their lifetime and have presently ceased smoking; and non-smokers reported having smoked <100 cigarettes during their lifetimes, as previously performed [15].

### 2.3. Data Management, Analysis and Statistical Methods

All datasets from the NHANES in 2001–2002 to 2017–2018 were analyzed through IBM SPSS Statistics version 26.0.0.0 for Macintosh (IBM Corp., Armonk, NY, USA), and data was uploaded through SAS Universal Viewer and handled with Microsoft Office Excel. Descriptive measures were outlined through mean ± standard deviation (SD) for continuous variables and number of cases (n) and percentage (%) for categorical variables. After inspection of data normality and homoscedasticity, the Mann–Whitney test was used to compare the continuous variables. The Chi-square test was used to compare the categorical variables. Multivariate linear regression analyses were used to model the influence of the number of missing teeth on SBP, DBP and Hba1c levels. Beta estimates (B) and correspondent standard errors (SE) were calculated for different adjustment levels. The model adjustment was made progressively by including Hba1c (%) (for the SBP or DBP models)/SBP (for the Hba1c model), smoking habits, education level and age, respectively. A significance level of 5% was set in all inferential analyses.

## 3. Results

### Characteristics of Participants

From an initial pool of 155,729 participants, 382 participants had reported medication indicative of PD and received a dental examination. Lastly, 214 met our inclusion criteria and were included in the analysis (Figure 1).

Our sample presented an equal number of female and male counterparts (Table 1). Upon comparison of this sample according to sex, females presented a lower age range (*p* = 0.008) and a significant difference in the distribution of marital status (*p* = 0.013). However, we found no statistically significant differences between females and males on other sociodemographic parameters (such as race/ethnicity, educational level or family income-to-poverty ratio) or even on the evaluated systemic health parameters. The majority of participants were Non-Hispanic White (65.4%), reported higher education (48.1%) and had never smoked (50.5%). DM and high blood pressure cases showed no statistical differences among sexes. Dental parameters were also assessed and compared according to sex (Table 1). Our overall sample presented approximately 9.7 teeth missing excluding wisdom teeth, with 23.8% presenting severe tooth loss and 18.2% being fully edentulous.

Additionally, we then compared the participants’ characteristics according to the severity of tooth loss (Table 2). Patients with severe tooth loss were significantly older (71.6, *p* < 0.001), and the majority reported education levels below high school (43.1%). Oppositely, most patients with non-severe tooth loss reported higher education levels (52.1%, *p* = 0.045). Furthermore, there was also a significant difference in the distribution of marital status according to the severity of tooth loss (*p* = 0.006). Regarding smoking status, whilst most participants with non-severe tooth loss reported never having smoked, those with severe tooth loss mostly reported being active smokers (55.8% vs. 45.1%, *p* = 0.008).

In addition, severe tooth loss cases reported a higher mean value of chronic medical conditions (2.6 vs. 1.8, *p* = 0.012), a higher percentage of DM cases (29.4% vs. 16.0%, *p* = 0.002) and an overall higher mean value of glycated hemoglobin (6.1 vs. 5.8, *p* = 0.001). Finally, the mean value of SBP and DBP were found to be significantly different between PD individuals, with severe tooth loss and without (*p* = 0.015 and *p* < 0.001, respectively).

To assess whether the number of missing teeth impacted both measures of BP, we carried out crude and adjusted linear regression analyses for SBP and DBP (Table 3). In what concerns SBP, the crude model revealed a significant relationship with the number of missing teeth (B = 0.34, SE = 0.14, *p* < 0.05). Notably, the fully adjusted analysis confirmed a steady significance (Models 2–4, Table 3), but not statistically significant relationship when age was included in Model 5 (B = 0.10, SE = 0.16, *p* > 0.05). Concerning DBP, the crude model demonstrated a relationship with the number of missing teeth (B = 0.26, SE = 0.09, *p* < 0.01). In the same way, the fully adjusted analysis confirmed a steady significance (Models 2–4, Table 3), however when age was included in Model 5, this statistical significance was lost (B = 0.16, SE = 0.10, *p* > 0.05).

Lastly, when assessing the relationship between Hba1c levels and the number of missing teeth through a linear model, neither the crude nor adjusted models revealed significance (*p* > 0.05) (Table 4).

## 4. Discussion

The results acquired in the present study from the NHANES database report a significant level of tooth loss in PD patients, as 23.8% presented severe tooth loss and 18.2% were fully edentulous. Additionally, BP levels in PD individuals showed an association with number of missing teeth and were confounded by the age factor. However, missing teeth and Hba1c revealed no association. These results might be relevant considering the lack of oral data regarding PD patients, the apparent oral care needs these patients present and the possible systemic repercussions.

In fact, severe tooth loss in individuals with PD occurred not only at older age groups (71.6, *p* < 0.001), but was also predictably associated with lower education levels and active smoking. Our results align with existing literature, as advanced age, lower income levels, less education and smoking are amongst the known risk factors for edentulism [45,46]. This sample of participants was evenly composed regarding sex and no significant differences were found on most sociodemographic and systemic health parameters.

The motor and cognitive impairments of PD ultimately compromise everyday life activities, such as the efficacy to carry out adequate oral hygiene habits [5,7,9]. Furthermore, the deterioration of oral status may also suffer from the medication regimen most PD patients undertake as the quantity and quality of secreted saliva change, contributing to unpleasant oral conditions and/or its progression [12,47].

Regarding the interplay with systemic health, severe tooth loss cases reported the concomitant presence of various chronic medical conditions, as well as a higher prevalence of DM cases (29.4% vs. 16.0%, *p* = 0.002). Tooth loss is the ultimate outcome of both periodontitis [48,49,50,51] and carious lesions [52]. Moreover, we recently demonstrated that PD patients are at higher risk of presenting periodontitis [53] and tooth loss is an accepted indicator of past or present periodontal disease (as previously reported) [54]. This increased risk PD patients present for periodontitis might be due to the fine motor handicaps and cognitive deficits present in later disease stages that ultimately compromise daily oral hygiene habits and potentiate periodontal issues [9]. Importantly, the high prevalence of chronic systemic conditions observed in severe tooth loss is noteworthy, and although we cannot infer causality, this should be clarified in further research. Moreover, edentulism has been shown to be a potential risk indicator for the onset of DM, mainly because dietary changes occur by impaired mastication due to missing teeth [55].

When it comes to the association of missing teeth with systemic health parameters in PD patients, a positive association was found with BP levels, as previously reported [34,56,57]. However, there was no significant data supporting an association between Hba1c and the number of missing teeth, contradicting literature on the edentulism–DM link abovementioned [55]. A possible reason might be the limited sample size, but this warrants further clarification.

Research shows a positive association between hypertension and higher levels of tooth loss, although the used sample only appraised people under 65 years of age [30,31]. Knowing that the late onset form is most common in PD, usually setting off around 65–70 years of age [1,6], the hypertension-tooth loss link should be confirmed in higher age groups as to determine the true age role in such relation.

All in all, the present study presents strengths and limitations to be appraised. Firstly, the study sample was acquired from a large representative U.S. population survey, from a vast collection of datasets through an 18-year time span (2001–2018). Additionally, our research aimed to evaluate a previously proven association between BP and tooth count [34], but this time in a group of patients taking PD-based medication. Notwithstanding and despite the reproducibility of the study design, the final included sample was relatively small. This might be justified with the fact that PD prevalence varies from 0.1% to 0.2% in unselected populations at any time [1]. Furthermore, the method employed to categorize a PD case is debatable, although this method has been previously accepted [15]. Moreover, a consistent report of oral health data was lacking as the oral health examination protocols varied across NHANES waves, which limited the extent of our research to the analysis of tooth count. For this reason, assessing periodontitis and caries was not possible in the overall sample, and this should be considered in future research. In addition, the conducted analysis was limited to the correlation of tooth loss in PD patients with HBP and Hba1c levels. This was dependent on the available laboratory data from the NHANES database, which was also not fully consistent throughout all waves (from 2001 to 2018).

Therefore, the novelty of this study is the search for the link between BP and Hba1c levels and tooth loss count in PD cases, which has not been explored to date. Future studies should further explore a possible association between tooth loss, periodontitis and caries and overall oral health status of PD patients with other systemic markers of interest.

## 5. Conclusions

Severe tooth loss seems to be significantly prevalent among people with PD. The number of missing teeth in PD individuals presented a positive linear relationship with BP levels, even though age constituted a confounding factor. There was no evidence of a relationship between the number of missing teeth and Hba1c levels in PD patients. Future studies should further analyze the oral health repercussions of PD patients on systemic health and well-being.

## Figures and Tables

**Figure 1 ijerph-18-05032-f001:**
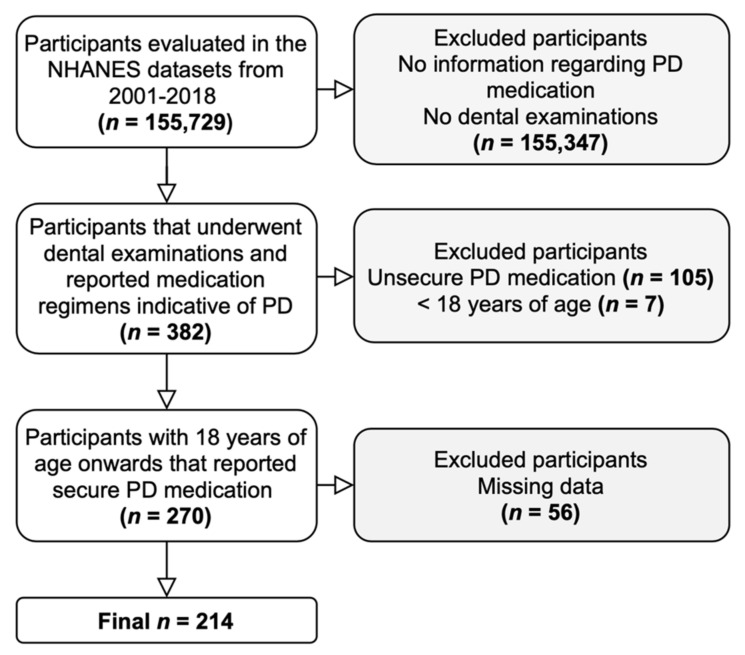
Flowchart of participants.

**Table 1 ijerph-18-05032-t001:** General characteristics of study participants according to gender.

Variable	Females (n = 107)	Males (n = 107)	*p*-Value *	Overall (n = 214)
**Age (years), mean (SD)**	60.5 (16.8)	66.3 (15.6)	0.008	63.4 (16.5)
**Race/ethnicity, n (%)**				
Mexican American	9 (8.4)	11 (10.3)	0.239	20 (9.4)
Other Hispanic	7 (6.5)	9 (8.4)		16 (7.5)
Non-Hispanic White	69 (64.5)	71 (66.4)		140 (65.4)
Non-Hispanic Black	17 (15.9)	16 (15.0)		33 (15.4)
Other Race—Including Multi-Racial	5 (4.7)	0 (0.0)		5 (2.3)
**Education level, n (%)**				
<High school	28 (26.2)	36 (33.6)	0.306	64 (29.9)
High school	22 (20.6)	25 (23.4)		47 (22.0)
>High school	57 (53.3)	46 (43.0)		103 (48.1)
**Marital Status, n (%)**				
Single	16 (15.0)	16 (15.0)	0.013	32 (15.0)
Married/Living with a partner	37 (34.6)	59 (55.1)		96 (44.9)
Divorced/Separated/Widowed	54 (50.5)	32 (29.9)		86 (40.2)
**Family income/poverty ratio, mean (SD)**	2.1 (1.5)	2.26 (1.5)	0.611	2.19 (1.5)
**Smoking status, n (%)**				
Non-smokers	56 (52.3)	52 (48.6)	0.365	108 (50.5)
Former-smokers	24 (22.4)	19 (17.8)		43 (20.1)
Active-smokers	27 (25.2)	36 (33.6)		63 (29.4)
**No. of chronic medical conditions, mean (SD)**	1.9 (1.6)	2.0 (2.0)	0.632	2.0 (1.8)
**Diabetes Mellitus, n (%)**	18 (16.8)	23 (21.5)	0.212	41 (19.2)
**Uncontrolled diabetic patients (Hba1c > 8%), n (%)**	3 (2.8)	5 (4.7)		8 (3.7)
**Hba1c, mean (SD)**	5.8 (0.9)	5.9 (1.0)	0.503	5.9 (0.9)
**Hypertension, n (%)**	68 (81.3)	62 (82.2)	1.000	109 (50.9)
**Uncontrolled hypertension, n (%)**	65 (95.6)	55 (88.7)	0.239	67 (31.3)
**SBP, mean (SD)**	129.6 (25.5)	128.1 (14.4)	0.935	128.8 (20.5)
**DBP, mean (SD)**	69.06 (12.2)	69.94 (13.8)	0.504	69.51 (12.9)
**Dental examination**				
**Edentulism, n (%)**	21 (19.6)	18 (16.8)	0.723	39 (18.2)
**Severe tooth loss (<9 teeth present), n (%)**	25 (23.4)	26 (24.3)	0.664	51 (23.8)
**Total teeth missing excluding wisdom teeth, mean (SD)**	9.4 (10.4)	10.0 (10.5)	0.520	9.7 (10.4)

DBP—Diastolic Blood Pressure; Hba1c—Hemoglobin A1C level; n—number of cases; SBP—Systolic Blood Pressure; SD—Standard Deviation. *Mann–Whitney test for continuous variables and Chi-square test for categorical variables.

**Table 2 ijerph-18-05032-t002:** General characteristics of study participants according to the severity of tooth loss.

Variable	Severe Tooth Loss (n = 51)	Non-Severe Tooth Loss (n = 163)	*p*-Value*
**Age (years), mean (SD)**	71.6 (11.3)	60.8 (17.0)	<0.001
**Race/ethnicity, n (%)**			
Mexican American	4 (7.8)	16 (9.8)	0.058
Other Hispanic	8 (15.7)	8 (4.9)	
Non-Hispanic White	34 (66.7)	106 (65.0)	
Non-Hispanic Black	5 (9.8)	28 (17.2)	
Other Race—Including Multi-Racial	0 (0)	5 (3.1)	
**Education level, n (%)**			
<High school	22 (43.1)	42 (25.8)	0.045
High school	11 (21.6)	36 (22.1)	
>High school	18 (35.3)	85 (52.1)	
**Marital Status, n (%)**			
Single	2 (3.9)	30 (18.4)	0.006
Married/Living with a partner	23 (45.1)	73 (44.8)	
Divorced/Separated/Widowed	26 (51.0)	60 (36.8)	
**Family income/poverty ratio, mean (SD)**	2.0 (1.4)	2.2 (1.5)	0.408
**Smoking status, n (%)**			
Non-smokers	17 (33.3)	91 (55.8)	0.008
Former-smokers	11 (21.6)	32 (19.6)	
Active-smokers	23 (45.1)	40 (24.5)	
**Chronic medical conditions, mean (SD)**	2.6 (2.1)	1.8 (1.6)	0.012
**Diabetes, n (%)**	15 (29.4)	26 (16.0)	0.002
**Uncontrolled diabetic patients (Hba1c > 8%), n (%)**	1 (6.7)	5 (3.1)	0.649
**Hba1c, mean (SD)**	6.1 (0.8)	5.8 (1.0)	0.001
**Hypertension, n (%)**	33 (64.7)	97 (59.5)	0.456
**Uncontrolled hypertension, n (%)**	31 (60.8)	89 (54.6)	0.736
**SBP, mean (SD)**	135.0 (25.5)	128.0 (20.8)	0.015
**DBP, mean (SD)**	64.0 (14.7)	70.9 (13.1)	<0.001

DBP—Diastolic Blood Pressure; Hba1c—Hemoglobin A1C level; n—number of cases; SBP—Systolic Blood Pressure; SD—Standard Deviation. *Mann–Whitney test for continuous variables and Chi-square test for categorical variables.

**Table 3 ijerph-18-05032-t003:** Crude and adjusted linear regression models for the number of missing teeth towards Systolic Blood Pressure (SBP) and Diastolic Blood Pressure (DBP). Values are presented as B estimates (Standard Error).

	SBP	DBP
Model 1	0.34 (0.14) *	0.26 (0.09) **
Model 2	0.33 (0.15) *	0.26 (0.09) **
Model 3	0.34 (0.15) *	0.28 (0.09) **
Model 4	0.30 (0.15) *	0.30 (0.09) **
Model 5	0.10 (0.16)	0.16 (0.10)

Model 1—Unadjusted model; Model 2—Includes adjustment for Hba1c level; Model 3—Includes adjustment for Hba1c level and smoking habits; Model 4—Includes adjustment for Hba1c level, smoking habits and education level; Model 5—Includes adjustment for Hba1c level, smoking habits, education level and age. * *p* < 0.05; ** *p* < 0.01.

**Table 4 ijerph-18-05032-t004:** Crude and adjusted linear regression models for number of missing teeth towards glycated hemoglobin (Hba1c) for the overall sample (N = 214). Values are presented as B estimates (Standard Error).

	Hba1c
Model 1	0.01 (0.01)
Model 2	0.01 (0.01)
Model 3	0.01 (0.01)
Model 4	0.01 (0.01)
Model 5	0.00 (0.01)

Model 1—Unadjusted model; Model 2—Includes adjustment for SBP; Model 3—Includes adjustment for SBP and smoking habits; Model 4—Includes adjustment for SBP, smoking habits and education level; Model 5—Includes adjustment for SBP, smoking habits, education level and age.

## Data Availability

Data available in a publicly accessible repository that does not issue DOIs. Publicly available datasets were analyzed in this study. This data can be found here: www.cdc.gov/nchs/nhanes.htm (accessed on 15 April 2021).

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
