# Peer review of "Tooth Loss and Blood Pressure in Parkinson’s Disease Patients: An Exploratory Study on NHANES Data"

_ijerph, 2021, doi:10.3390/ijerph18095032_

Round 1
Reviewer 1 Report
This is an interesting study.
The authors well explained the study design however the rationale of the study in the introduction should be deeply commented and previous studies on the topic should be described.
The statistical analysis should present an analysis on why including the age in the model there is a lost of statistically significance.
The author should stress more the importance and the clinical relevance of their findings.
Author Response
Dear Editors of the International Journal of Environmental Research and Public Health,
We are delighted to submit our revised manuscript titled “Tooth loss and blood pressure in Parkinson’s Disease patients: an exploratory study on NHANES data” (Manuscript ID ijerph-1205096).
All editorial and reviewers’ comments were considered. Please find appended a track-changes draft of the manuscript and a point-by-point rebuttal to all comments raised as detailed below. We hope our responses are now suitable for publication in the International Journal of Environmental Research and Public Health.
Reviewer 1
This is an interesting study.
Point 1: The authors well explained the study design however the rationale of the study in the introduction should be deeply commented and previous studies on the topic should be described.
Response 1: We thank and acknowledge your suggestion. Accordingly, we corrected our typo and previous studies on the topic were cited - “However, the link between BP and Hba1c measures and tooth loss count in PD cases has never been studied, although a prospective positive link may arise according to recent several lines of evidence [18,20,22,28,29].”. Furthermore, the rationale of the study in the introduction was further contextualized with the addition of the following information “High BP, which burdens about 45% of the US adult population, is associated with cognitive decline and premature mortality rates [15,16,17], and not only has been linked to periodontitis [18,19] and tooth loss [20-23], but also blood pressure fluctuations have been associated with the autonomic dysfunctions in PD [24]. Also, diabetes mellitus, affecting approximately 13% of US adults [25], has a known bidirectional relationship with periodontitis [26] and it is a possible risk factor to PD, although further research is needed to clarify the biological mechanisms of such association [27].”.
Point 2: The statistical analysis should present an analysis on why including the age in the model there is a lost of statistically significance.
Response 2: We appreciate this remark, however we have previously done this according to the past evidence discussed, mainly research from our research group (https://www.mdpi.com/1660-4601/18/1/285 OR https://www.mdpi.com/2077-0383/9/5/1585/htm#B34-jcm-09-01585).
Point 3: The author should stress more the importance and the clinical relevance of their findings.
Response 3: We added the following sentence to meet your suggestion - “Therefore, the novelty of this study is the search for the link between BP and Hba1c levels and tooth loss count in PD cases, which has never been explored to date.”.

Reviewer 2 Report
The manuscript submitted to IJERPH "Tooth loss and blood pressure in Parkinson’s Disease patients: an exploratory study on NHANES data" is an interesting paper that shows the hypothesis of PD could be related to tooth loss and other comorbidity.
The manuscript is well written, with good English.
I would like to congratulate the authors for the statistical in-depth analysis.
I have only some question and suggestion for the authors:
Methods
Could you add the code of the ethics committee?
- Health characteristics part
"Diabetes Mellitus (DM) and hypertension were separately appraised variables due to their known strong impact on oral health [25-27]."
I suggest that you refer to these interesting recent paper about the influence of DM, hypertension and other comorbidity on oral health.
After the implementation of the methods and discussion part, this manuscript must be revised and re-evaluated for the publication suitability.
I am available to a second round review.
Author Response
Dear Editors of the International Journal of Environmental Research and Public Health,
We are delighted to submit our revised manuscript titled “Tooth loss and blood pressure in Parkinson’s Disease patients: an exploratory study on NHANES data” (Manuscript ID ijerph-1205096).
All editorial and reviewers’ comments were considered. Please find appended a track-changes draft of the manuscript and a point-by-point rebuttal to all comments raised as detailed below. We hope our responses are now suitable for publication in the International Journal of Environmental Research and Public Health.
Reviewer 2
The manuscript submitted to IJERPH "Tooth loss and blood pressure in Parkinson’s Disease patients: an exploratory study on NHANES data" is an interesting paper that shows the hypothesis of PD could be related to tooth loss and other comorbidity.
The manuscript is well written, with good English.
Point 1: I would like to congratulate the authors for the statistical in-depth analysis.
Response 1: We deeply appreciate your detailed revision of our manuscript, as we consider the subject matter for this study an extremely relevant step towards understanding the link between oral health in PD patients and other systemic comorbidities.
Point 2: I have only some question and suggestion for the authors:
Methods
Could you add the code of the ethics committee?
Response 2: As requested, ethics committee codes for all analysed NHANES waves were added on the “Study Design and Participants” section of Materials and Methods - “Furthermore, all waves of NHANES were reviewed and approved by the Centers for Disease Control (CDC) and Prevention National Center for Health Statistics Research (NCHS) Research Ethics Review Board (Atlanta USA, protocols #98-12, #2005-06, #2011-17 and #2018-01), and all study participants provided written informed consent [22].”.
Point 3: Health characteristics part
"Diabetes Mellitus (DM) and hypertension were separately appraised variables due to their known strong impact on oral health [25-27]."
I suggest that you refer to these interesting recent paper about the influence of DM, hypertension and other comorbidity on oral health.
DOI:10.1007/s00784-020-03420-3
DOI: 10.1186/s12903-020-01219-y
Response 3: We thank and acknowledge the suggestion. We have cited the second suggested paper (DOI: 10.1186/s12903-020-01219-y), as with the first one there might have been a typo on the DOI, as it does not fully align with the scope of this study. We remain available to cite further papers you may find appropriate.
Point 4: Discussion I suggest to improve the discussion part about the risk of PD patients to present an higher grade of periodontitis
Response 4: We appreciate the remark and fully agree that the higher risk of periodontitis in PD patients should be addressed more clearly in the discussion section. Therefore, the following sentence was added “This increased risk PD patients present for periodontitis might be due to the fine motor handicaps and cognitive deficits present in later disease stages, that ultimately compromise daily oral hygiene habits and potentiate periodontal issues [8]”.
Point 5: After the implementation of the methods and discussion part, this manuscript must be revised and re-evaluated for the publication suitability.
Response 5: We hope to have met your concerns and remain totally available for further improvements.

Reviewer 3 Report
This manuscript evaluating tooth loss severity in Parkinson Disease patients and the impact of missing teeth on blood pressure and hbA1c, indeed falls within the scope of the journal. The manuscript is well written and fairly conducted in general. Below are some comments: 1-Line 83-87:"In this sense, 83 PD cases were defined if patients reported to use any of the indicated medication: Benztro- 84 pine, Carbidopa, Levodopa, Ropinirole, Methyldopa, Entacapone, and Amantadine. 85 Then, PD cases were selected through the report of secure PD medications (Benztropine, 86 Carbidopa, Levodopa, Ropinirole, Methyldopa, Entacapone and Amantadine)"
The two sentences should be combined in one to avoid repetition.
2-Line 120: Categories of smokers varies between the body of the manuscript (active, former, Non smoker) vs table (never ,former, current).Please use consistentTerminology. 3-Line 154: "
DM and high blood pressure cases were evenly distributed 154 among sexes."
DM and HBP are mentioned to be evenly distributed, however the numbers on the table doesn't support that. Can you please check that.
Please add in the introduction prevalence of Parkinson disease, tooth loss due to periodontal disease or caries, diabetes and hypertension in USA population.
Also discuss briefly biologic plausibility underlying the correlation between Diabetes , high blood pressure, tooth loss with Parkinson Disease.
Author Response
Dear Editors of the International Journal of Environmental Research and Public Health,
We are delighted to submit our revised manuscript titled “Tooth loss and blood pressure in Parkinson’s Disease patients: an exploratory study on NHANES data” (Manuscript ID ijerph-1205096).
All editorial and reviewers’ comments were considered. Please find appended a track-changes draft of the manuscript and a point-by-point rebuttal to all comments raised as detailed below. We hope our responses are now suitable for publication in the International Journal of Environmental Research and Public Health.
Reviewer 3
This manuscript evaluating tooth loss severity in Parkinson Disease patients and the impact of missing teeth on blood pressure and hbA1c, indeed falls within the scope of the journal. The manuscript is well written and fairly conducted in general. Below are some comments:
Point 1: 1-Line 83-87:
"In this sense, 83 PD cases were defined if patients reported to use any of the indicated medication: Benztro- 84 pine, Carbidopa, Levodopa, Ropinirole, Methyldopa, Entacapone, and Amantadine. 85 Then, PD cases were selected through the report of secure PD medications (Benztropine, 86 Carbidopa, Levodopa, Ropinirole, Methyldopa, Entacapone and Amantadine)"
The two sentences should be combined in one to avoid repetition.
Response 1: We agree with your comment and apologise for the writing lapse. The abovementioned sentences have been reformulated into one: “In this sense, PD cases were defined if patients reported the use of the following secure PD medications: Benztropine, Carbidopa, Levodopa, Ropinirole, Methyldopa, Entacapone, and Amantadine [13].”.
Point 2: 2-Line 120: Categories of smokers varies between the body of the manuscript (active, former, Non smoker) vs table (never ,former, current). Please use consistent Terminology.
Response 2: Thank you for the detailed review. We apologise for the typo and have corrected the terminology regarding smoking status throughout the manuscript.
Point 3: 3-Line 154: "DM and high blood pressure cases were evenly distributed 154 among sexes." DM and HBP are mentioned to be evenly distributed, however the numbers on the table doesn't support that. Can you please check that.
Response 3: After carefully re-evaluating data, we agree that this sentence was susceptible to misinterpretations. Thus, we proceeded to correct it, and now reads: “DM and high blood pressure cases showed no statistical differences among sexes.”. We thank you for bringing this to our attention.
Point 4: Please add in the introduction prevalence of Parkinson disease, tooth loss due to periodontal disease or caries, diabetes and hypertension in USA population.
Response 4: As requested, the prevalence of PD, diabetes and hypertension in the USA population have been added in the introduction section, as we fully agree it is important contextualization data to the conditions we associate in our study - “Parkinson’s Disease (PD) is a slowly progressive and over time disabling condition of the nervous system, affecting around 680000 of US citizens and less than 1% of north Americans over 45 years of age [1,2].”, “High BP, which burdens about 45% of the US adult population, …” and “Also, diabetes mellitus, affecting approximately 13% of US adults [25], …”. However, the cause of tooth loss due to periodontal disease or caries was not possible to add due to the scarcity of literature regarding the USA.
Point 5: Also discuss briefly biologic plausibility underlying the correlation between Diabetes , high blood pressure, tooth loss with Parkinson Disease.
Response 5: The biologic plausibility underlying the correlation between tooth loss with PD was explored in the introduction with the following sentences - “The oral health condition of PD patients tends to be deteriorated due to the motor and cognitive dysfunction this neurodegenerative condition presents [8,9]. As a consequence, individuals with PD have deficient oral hygiene which precipitates the development of periodontitis, carious lesions, and consequently retained roots [10] and tooth loss [11].”. Also, we completely agree with the suggestion to further explain the diabetes-PD link and the HBP-PD link. Accordingly, the following information was added “High BP, which burdens about 45% of the US adult population, is associated with cognitive decline and premature mortality rates [15,16,17], and not only has been linked to periodontitis [18,19] and tooth loss [20-23], but also blood pressure fluctuations have been associated with the autonomic dysfunctions in PD [24]. Also, diabetes mellitus, affecting approximately 13% of US adults [25], has a known bidirectional relationship with periodontitis [26] and it is a possible risk factor to PD, although further research is needed to clarify the biological mechanisms of such association [27].”.

Round 2
Reviewer 2 Report
The authors followed the suggestions.
The manuscript could be improved in several parts, increasing readers' interesting.
I suggest to correlate DM and hypertension with other pathologies and the influence on the oral health.
Author Response
Dear Editors of the International Journal of Environmental Research and Public Health,
We are delighted to re-submit our revised manuscript titled “Tooth loss and blood pressure in Parkinson’s Disease patients: an exploratory study on NHANES data” (Manuscript ID ijerph-1205096).
All editorial and reviewers’ comments were considered. Please find appended a track-changes draft of the manuscript and a point-by-point rebuttal to all comments raised as detailed below. We hope our responses are now suitable for publication in the International Journal of Environmental Research and Public Health.
"The authors followed the suggestions.
Point 1: The manuscript could be improved in several parts, increasing readers' interesting.
I suggest to correlate DM and hypertension with other pathologies and the influence on the oral health."
Response 1: We kindly thank and acknowledge your suggestion. Accordingly, we have further explained the correlation of DM with other pathologies, including its influence on oral health, by reformulating the following sentence of the introduction “Also, diabetes mellitus, affecting approximately 13% of US adults [26], and whose cardiovascular, renal, ocular and neuronal complications are well established in literature [27], has also a known bidirectional relationship with severe periodontitis [28] and it is a possible risk factor to PD, although further research is needed to clarify the biological mechanisms of such association [29].” Furthermore, we have clarified the term “periodontitis” at the site of first mentioning - “As a consequence, individuals with PD have deficient oral hygiene which precipitates the development of periodontitis (a chronic inflammatory condition of the supporting structure of the teeth) [10], carious lesions, and consequently retained roots [11] and tooth loss [12].” - which has been used throughout the manuscript, and heavily exemplifies the burden of DM, hypertension and even PD in oral health. Also, hypertension has been correlated with other pathologies (including those of the oral cavity) in the following sentence “High BP, which burdens about 45% of the US adult population, is associated with cognitive decline and premature mortality rates [16,17,18], and not only has been linked to periodontitis [19,20] and tooth loss [21-24], but also blood pressure fluctuations have been associated with the autonomic dysfunctions in PD [25].”.
Additionally, to clarify the focus of our study on the association between tooth count in PD patients with their HBP and Hba1c levels, we added the following information in the discussion section - “Also, the conducted analysis was limited to the correlation of tooth loss in PD patients with HBP and Hba1c levels. This was dependable on the available laboratory data from the NHANES database, which was also not fully consistent throughout all waves (from 2001 to 2018). Therefore, the novelty of this study is the search for the link between BP and Hba1c levels and tooth loss count in PD cases, which has never been explored to date. Future studies should further explore a possible association between tooth loss, periodontitis and caries and overall oral health status of PD patients with other systemic markers of interest.”.
